# Anti-phage defence through inhibition of virion assembly

Pramalkumar H. Patel [1], Véronique L. Taylor [1], Chi Zhang[2], Landon J. Getz [1], Alexa D. Fitzpatrick[1], Alan R. Davidson [1,2] & Karen L. Maxwell [1] ✉

Bacteria have evolved diverse antiviral defence mechanisms to protect themselves against phage infection. Phages integrated into bacterial chromosomes, known as prophages, also encode defences that protect the bacterial hosts in which they reside. Here, we identify a type of anti-phage defence that interferes with the virion assembly pathway of invading phages. The protein that mediates this defence, which we call Tab (for 'Tail assembly blocker'), is constitutively expressed from a *Pseudomonas aeruginosa* prophage. Tab allows the invading phage replication cycle to proceed, but blocks assembly of the phage tail, thus preventing formation of infectious virions. While the infected cell dies through the activity of the replicating phage lysis proteins, there is no release of infectious phage progeny, and the bacterial community is thereby protected from a phage epidemic. Prophages expressing Tab are not inhibited during their own lytic cycle because they express a counter-defence protein that interferes with Tab function. Thus, our work reveals an anti-phage defence that operates by blocking virion assembly, thereby both preventing formation of phage progeny and allowing destruction of the infected cell due to expression of phage lysis genes.

Bacteria encode a variety of sophisticated defence systems to protect themselves against phage infection[1–6]. Temperate phages, which can exist within bacterial cells in a form known as prophages, have also been shown to encode diverse anti-phage defences that protect their host cell from competing phages in the environment[6–14]. These defences are not only found in intact prophages, but also cryptic prophages[15], satellite phages like P4[12], and phage-inducible chromosomal islands (PICIs)[16,17], and they have been shown to play important roles in phage-phage competition and bacterial survival[11,13,18–22].

Studies of bacterial and prophage-encoded anti-phage defence systems have uncovered a wide range of mechanisms of activity that can be broadly classified according to the stage of phage infection that they inhibit. Superinfection exclusion systems function at the cell surface, preventing phages from binding to their receptor[23–26] or inhibiting their ability to inject their genome across the cell envelope[20,27,28]. Other systems prevent phage replication after the phage genome enters the cell. These intracellular defences can provide direct immunity to save the infected cell, or they can promote or allow cell death though abortive infection[29,30]. Systems that target early steps and provide direct immunity include CRISPR-Cas[31] and restriction-modification[32], which recognize the injected phage DNA as foreign and destroy it; thus, preventing phage replication and ensuring cell survival. Some systems, such as CBASS[33], CapRel[34] and Avs[35], sense phage infection or detect a phage protein, and subsequently employ various types of toxic proteins or domains to mediate cell dormancy or death. These systems, often referred to as "abortive infection", result in the death or growth inhibition of infected cells so that no newly replicated phages are released[29]. A third type of intracellular defence is provided by systems that inhibit replication of the phage, but cell death occurs through the activity of a phage protein. Examples include the toxin-antitoxin defence systems ToxIN, where phage-induced shutoff of host transcription arrests growth[36] and DarTG, where phage-induced chromosome degradation is the likely cause of cell death[37].

[1]Department of Biochemistry, University of Toronto, Toronto, ON, Canada. [2]Department of Molecular Genetics, University of Toronto, Toronto, ON, Canada. ✉e-mail: karen.maxwell@utoronto.ca

In this work we report a novel prophage-encoded defence system that functions by inhibiting the tail assembly pathway of competing phages. This system, found in *Pseudomonas aeruginosa* phage JBD26, is constitutively expressed from the prophage and confers effective population level defence. The protein, known as Tab for Tail assembly blocker, recognizes the tape measure protein produced during the replication cycle of invading phages and inhibits tail assembly. This results in the release of defective phage intermediates lacking tails that are unable to go on to infect other cells in the community. Phages that encode Tab also possess a counter-defence protein that acts during their own lytic cycle to prevent self-targeting. This work characterizes a defence mechanism that directly targets and inhibits the virion assembly pathway, and thus provides a new archetype of anti-phage defence.

## Results

In a previous study, we identified a protein, referred to as Tab, that mediated strong anti-phage defence when expressed from the host genome-integrated form (prophage) of *Pseudomonas aeruginosa* phage JBD26[14]. This gene is classified as a "moron" as it adds "more on" the phage genome when it is present[38,39]. Like many other previously characterized morons, it provides the host cell with increased fitness by endowing phage resistance, thereby also increasing the fitness of the prophage. This 83-residue protein (YP_010299225), which is encoded in the phage late gene operon between the small and large terminase genes, shares no sequence similarity with any protein or protein domain of known function. To identify a group of phages that were sensitive to Tab activity, we challenged *P. aeruginosa* strain PA14 expressing Tab from a plasmid with a panel of phages that include members of the *Casadabanvirus* and *Beetrevirus* families, as well as several unclassified *Caudoviricetes* (Supplementary Table 1). We identified five phages that showed >1000-fold decrease in efficiency of plating in the presence of Tab, showing that this protein provides strong defence (Fig. 1a). These five phages all belong to the *Casadabanvirus* family and are closely related to phages MP22 and MP29, sharing a common genomic organization that encodes proteins with pairwise amino acid sequence identities of 80–90% in most cases[13]. Within the group of phages tested, only JBD26 and JBD24 encode a Tab homologue (Fig. 1b), and they were unaffected by its activity.

### Tab mediates defence by inhibiting phage tail assembly

To elucidate the mechanism of Tab function, we sought to determine which step in the phage lifecycle was arrested in the presence of this protein. To this end, we used Southern blots to monitor phage DMS3 genome replication over time. Within 15 min of infection, phage DNA began to accumulate in the cell and continued to increase markedly to the 75-min timepoint (Fig. 1c), which is the expected lifecycle time for

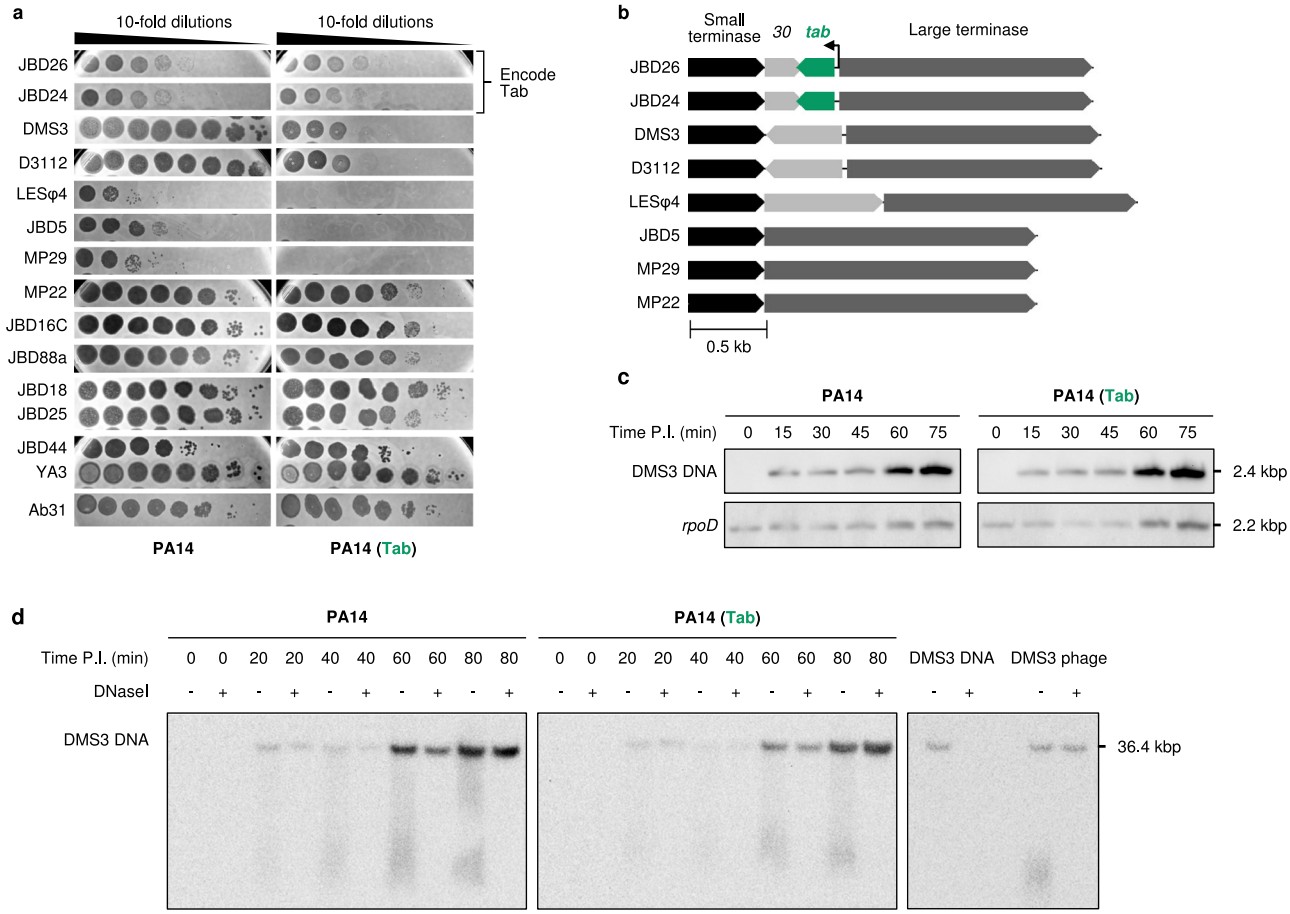

**Fig. 1 | Tab provides anti-phage defence. a** Phage infection assays. Ten-fold serial dilutions of indicated phages on PA14(ΔCRISPR) and PA14(ΔCRISPR) expressing Tab. Images shown are representative of three biological replicates. **b** Gene diagrams of the gene *31* (*tab*) region in related phages. The small and large terminase genes encode proteins that share >90% amino acid sequence identity among these phages. The arrow indicates the promoter of gene *31*. Scale bar, 500 base pairs. **c** Southern blot analysis assessing the replication and accumulation of DMS3 genomic DNA during phage infection in the absence and presence of Tab. DMS3 genomic DNA was detected using digoxigenin-labeled ssDNA probes. *RpoD* was used as a loading control. The data shown are representative of three biological replicates. Source data are provided. **d** Southern blot analysis assessing the accumulation of packaged DMS3 genomes during phage infection in the presence and absence of Tab. Purified DMS3 genomic DNA and mature DMS3 phage particles were used as controls (right). Time P.I., time post-infection. The data shown are representative of three biological replicates. Full-sized blots are presented in Supplementary Fig. 1a.

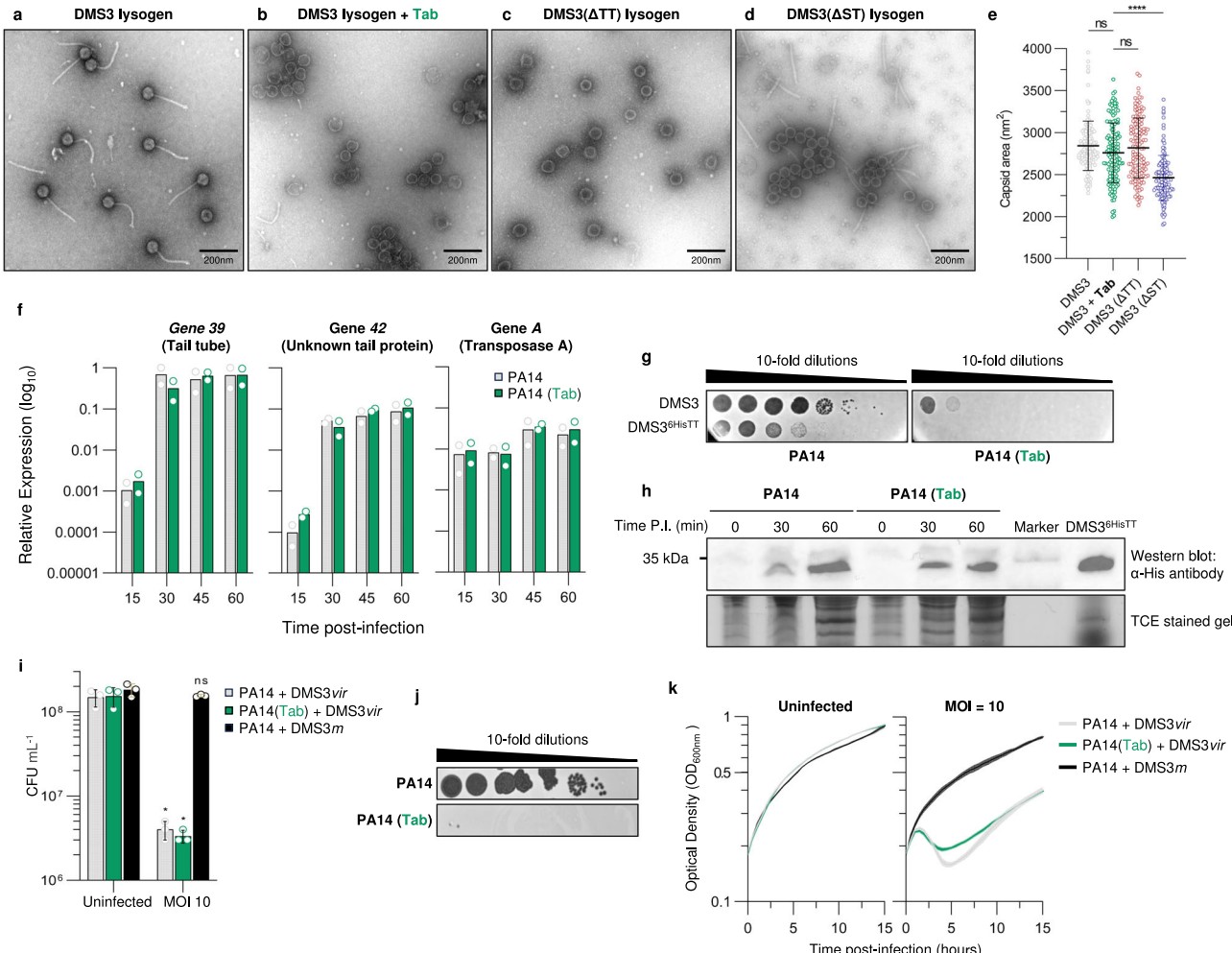

**Fig. 2 | Tab blocks phage tail assembly. a–d** Representative negatively stained transmission electron micrographs of phage particles and intermediates produced by DMS3 prophage induction in the absence (**a**) and presence (**b**) of Tab. Induction of DMS3 prophages containing the (**c**) tail tube (ΔTT), and (**d**) small terminase (ΔST) single gene deletions. Data are representative of three independent biological replicates. **e** Scatter plot of capsid area (nm²) for the DMS3 prophages described in panels (**a–d**) ($n = 135$ for each sample). The horizonal black line indicates the mean for each sample, and error bars indicate standard deviation. Statistical significance was measured using one-way ANOVA coupled with Dunnett's multiple comparison test using DMS3 with Tab as reference. Adjusted $p$-values are reported above the graph (not significant (ns), and ****$p < 0.0001$). $p$-values = 0.0838 (DMS3 + Tab vs. DMS3; ns), 0.3030 (DMS3 + Tab vs. DMS3 (ΔTT); ns), and « 0.0001 (DMS3 + Tab vs. DMS3 (ΔST)) (**f**) RT-qPCR analysis of early- (*A*) and late-expressed (*39 and 42*) DMS3*vir* genes in the presence and absence of Tab. Bars indicate averaged relative gene expression (log10) normalized to *rpoD* copy number for two biological replicates. **g** Phage plating assay of DMS3^6HisTT in the presence of Tab. Data are representative of three biological replicates. **h** Western blot analysis of 6-His-tagged tail tube production during infection in the presence and absence of Tab (upper panel) and the corresponding 2,2,2-Trichloroethanol (TCE) stained gel showing total protein load (lower panel). Data shown are representative of three biological replicates. Time P.I., time post-infection. **i** Cell survival assay (colony forming units; CFU mL⁻¹) in cells expressing Tab that were infected by DMS3*vir* or CRISPR-sensitive DMS3*m* at an MOI of 10. Uninfected cells are shown as a control. Bars represent mean values, with the error bars representing the standard deviation ($n = 3$ biologically independent samples). Asterisks show statistically significant differences relative to uninfected cells. *$p$-value < 0.05 (Paired $t$-test). $p$-values = 0.0171 for PA14, 0.0226 for PA14 (Tab), and 0.3035 for PA14 with DMS3*m*. **j** DMS3*vir* phage production in the absence or presence of Tab. Data are representative of three biological replicates. **k** Growth curves of PA14 with and without Tab and when challenged with DMS3 at an MOI of 10. Data are presented as mean values with shaded areas representing the standard deviation ($n = 3$ biologically independent samples). Source data are provided for panels (**e, f, h, I, k**).

this type of phage[40]. Importantly, we found that the level of DMS3 genomic DNA replication throughout the infection cycle was unaffected by the presence of Tab. This result demonstrates that Tab does not mediate destruction of the phage genome or prevent phage DNA replication; thus, it must block a later step in the phage lifecycle.

Phage virion assembly proceeds in an ordered manner; an empty prohead assembles, the phage genome is packaged into this prohead through the activity of the small and large terminase subunits, the packaged genome is stabilized within the head by the addition of head-tail joining proteins, and then tails, which are assembled in an independent pathway, are attached to the heads to form the mature phage particle[41,42]. Once the phage genome is packaged into the head, it becomes highly resistant to digestion by DNase I[43]. With this

knowledge, we infected cells with phage DMS3 and treated the resulting lysates with DNase I. A Southern blot of phage lysates produced in the presence and absence of Tab showed that comparable levels of packaged phage DNA were produced under both conditions (Fig. 1d, Supplementary Fig. 1a), implying that Tab does not inhibit the function of proteins required for head morphogenesis, or the DNA packaging process.

To further understand the mechanistic basis of anti-phage defence provided by Tab, we examined phage lysates using negative-stain transmission electron microscopy (TEM). Induction of a DMS3 prophage in the absence of Tab produced a lysate containing fully assembled virions, as expected (Fig. 2a). By contrast, lysates produced by DMS3 prophage induction carried out in the presence of Tab

displayed an abundance of empty heads, no tails, and few complete phage particles (Fig. 2b). Plating assays confirmed that the production of infectious phage particles was inhibited more than 100-fold in these conditions (Supplementary Fig. 1b, c). We also examined lysates produced by phage DMS3 infection of cells expressing Tab and found the same result – many empty heads and no assembled tails (Supplementary Fig. 1d).

The presence of empty heads in the lysates of phage DMS3 produced in cells expressing Tab was unexpected in light of our results showing that phage DNA was resistant to DNase I digestion when lysates were produced under the same conditions (Fig. 1d). To better understand exactly how the phage virion assembly pathway was being inhibited, we created two phage DMS3 deletion mutants – a small terminase mutant (ΔST) that cannot package the phage genome into the capsid, and a tail tube mutant (ΔTT) that lacks the major component of the phage tail and thus does not produce tails. We found that the lysate produced by the DMS3 mutant bearing a deletion in the gene encoding the tail tube protein lacked tails as expected, but also contained empty heads like those observed for wild type DMS3 produced in the presence of Tab (Fig. 2c). By contrast, the lysate of the mutant phage lacking its small terminase subunit contained empty proheads, as expected since DNA packaging cannot occur, and an abundant level of normal looking tails (Fig. 2d). Taken together, these results suggest that Tab blocks phage tail assembly. We surmise that the DNA-filled heads produced in the absence of tails are not fully stable and that preparation of the TEM grids causes DNA egress from the heads. This idea is supported by the morphology of the empty heads observed in the ΔTT mutant lysates (Fig. 2c). These empty heads contrast to those seen in the ΔST mutant lysate in being larger, thinner shelled and more varied in size and shape (Fig. 2b, c, d), indicating that DNA was packaged into these heads, head expansion occurred, and then the DNA fell out. This result is consistent with previous observations of an *E. coli* phage HK97 mutant defective for tail attachment[44].

To provide evidence that the DNA was being lost from the heads during our sample preparation, we performed two additional analyses. First, we subjected the lysates that we were preparing for TEM to DNase treatment before beginning the manipulations and showed that the DNA was protected inside the capsid at this point in both the phage produced in the presence of Tab and the tail tube mutant, but not the small terminase mutant (Supplementary Fig. 1e). Next, we used bioimaging analysis to calculate the average capsid sizes that were present in the different phage lysate TEM samples (Fig. 2e). We found that the average size of a DNA filled head from the wild-type phage infection was 2843 nm². This was similar the average size observed in both DMS3 + Tab (2759 nm²) and DMS3(ΔTT) (2818 nm²) samples (Fig. 2e). However, these samples showed less homogeneity in size than the wild-type phage heads, consistent with the more varied sizes and shapes noted in the electron micrographs. By contrast, the small terminase mutant displayed an average area of only 2462 nm², which is consistent with the smaller procapsid size known to exist before the phage genome is packaged into it. Together, these data provide evidence that the phage genome was packaged into the capsid in the presence of Tab and was subsequently lost.

The absence of tails observed in DMS3 infection in the presence of Tab could result from lack of expression of the tail proteins, or interference with phage tail assembly. To determine which of these two possibilities was correct, we examined tail gene transcription. Cells were infected in the presence or absence of Tab, and RT-qPCR was used to quantify transcript levels of two genes encoding protein components of the tail (TT protein, gene *39*; and tail protein of unknown function, gene *42*). We found that transcription of these two genes was detectable at 30 min and increased until 60 min, as expected for genes in the late operon (Fig. 2f, Supplementary Fig. 2). No differences in transcript levels were noted in the presence of Tab. Transcription of the early expressed transposase gene (*A*) and genes

involved in head formation (*29* and *33*) were also unaffected by the presence of Tab (Fig. 2f, Supplementary Fig. 2). To determine if translation of the tail tube protein also proceeded normally, we inserted DNA encoding a hexa-histidine (6-His) tag into the tail tube gene in the DMS3 phage genome in such a way that the tag was appended to the N-terminus of the tail tube protein (DMS3⁶ᴴⁱˢᵀᵀ). This mutant phage formed infectious particles that were still subject to Tab inhibition (Fig. 2g), and equal levels of 6-His tagged tail tube protein were observed in phage infections in the presence or absence of Tab (Fig. 2h, Supplementary Fig. 3). Taken together, these results support a model in which Tab specifically interferes with phage tail assembly.

Since Tab activity is mediated late in the phage life cycle, we expected that other late phage functions would also be expressed; in particular, the proteins responsible for cell lysis. To investigate this issue, we mixed phages and cells at a multiplicity of infection (MOI) of 10; after 15 min incubation, any unabsorbed phages were removed, and the cells were plated and grown overnight to assess survival rates. Cells initially infected by phages able to replicate and express the late gene operon do not go on to form colonies in this assay due to progression of the phage lysis pathway. We observed that in both the presence and absence of Tab, >97% of cells were killed by phage DMS3 (Fig. 2i). However, despite this rampant cell death, phage production was strongly inhibited in the presence of Tab (Fig. 2j). This situation contrasted with the immediate-acting defence provided by CRISPR-Cas, which mediated full protection of the cells from death by phage infection by the CRISPR-targeted phage mutant DMS3*m* (Fig. 2i). Monitoring these infection dynamics in liquid culture revealed that cell lysis occurred with similar timing in both the presence and absence of Tab (Fig. 2k), suggesting that the full phage life cycle was proceeding with similar timing. Thus, we believe that the phage lysis proteins are expressed in the presence of Tab and are the cause of death of the infected cells.

## Tab activity targets the phage tail tape measure protein
To identify the specific target of Tab activity, we attempted to isolate phage escape mutants using the method described by Stokar-Avihail et al. (2023)[45] and the liquid growth phage evolution experiments described by Srikant et al. (2022)[46]. However, these attempts were unsuccessful, suggesting that simple mutations in the phage are not sufficient to bypass this defence, or that mutations that are able to bypass this defence are lethal to the phage. As the sequences of tail proteins of phages that were inhibited by Tab are very similar in several phages that were not affected by its activity, we performed sequence comparisons to ascertain why some phages were targeted. We noted that the proteins that comprise the tail tip (encoded by genes *42-48*) generally showed high sequence conservation among phages that were both affected and unaffected by Tab activity (Fig. 3a, b). By contrast, the tail tube, tail assembly chaperone and the tape measure (TMP) proteins showed lower levels of sequence conservation, suggesting that these might be the target of Tab activity. To determine if one of these proteins was associated with the observed inhibition, we created a hybrid DMS3 phage where the gene encoding the TMP was replaced by the TMP gene found in phage JBD16C, which is not blocked by Tab activity. This hybrid phage, DMS3¹⁶ᶜ⁻ᵀᴹᴾ, formed infectious phage particles and was able to plate on cells expressing Tab, forming tiny plaques with less than a 10-fold decrease in plating efficiency, as compared to >10⁵-fold decrease observed for wild type phage DMS3 in the infection assay (Fig. 3c). This suggested that the TMP protein sequence is targeted by Tab activity. While creating this hybrid, we isolated a second DMS3 phage that had only the first 856 residues of the TMP replaced by the JBD16C sequence. This phage, DMS3¹⁶ᶜ⁻ᵀᴹᴾ⁸⁵⁶, was still highly susceptible to Tab mediated inhibition (Fig. 3c), demonstrating that protein sequence at the C-terminus of the TMP is linked with the inhibition of phage replication observed in the presence of Tab (Supplementary Fig. 4). As the TMP is a key component of

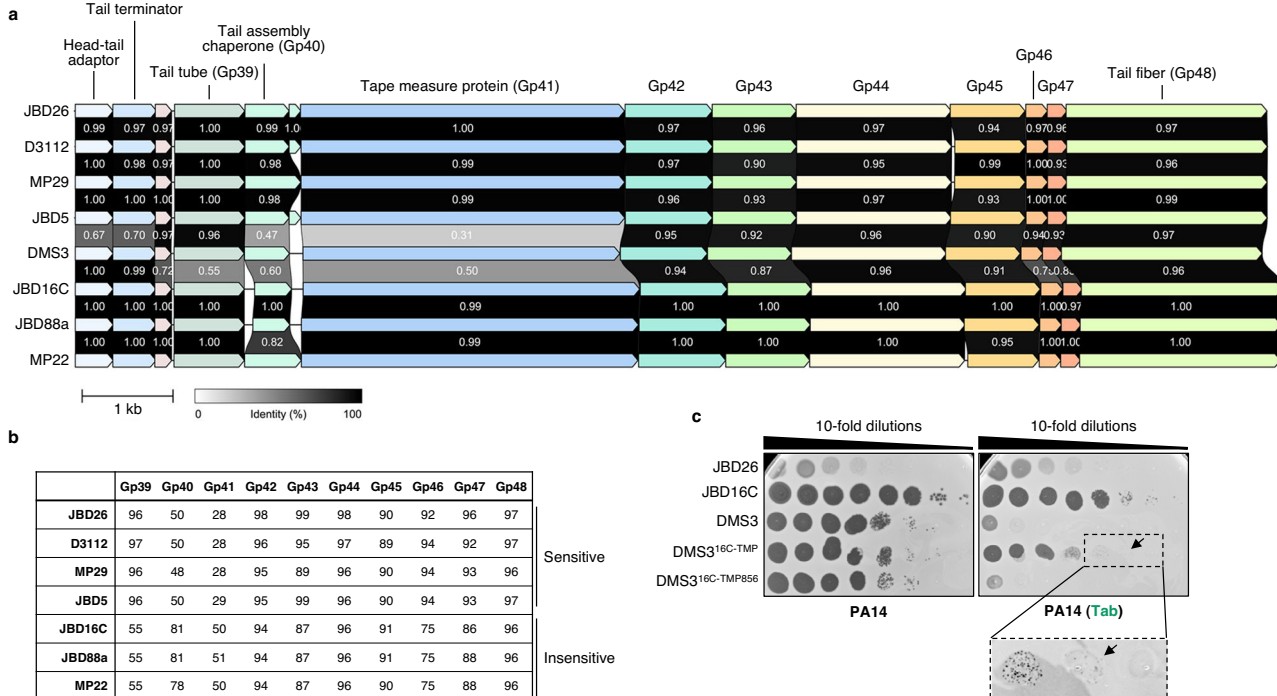

**Fig. 3 | Tab targets the tape measure protein (TMP). a** Diagram showing tail gene operon synteny and corresponding protein sequence conservation between phages sensitive or insensitive to Tab activity. The gene product (Gp) labels correspond to the gene number in phage DMS3. Genes encoding proteins sharing >30% amino acid sequence identity are colored the same. The numbers between phage operons indicate the percent shared protein sequence identity. Scale bar, 1000 base pairs. **b** A matrix highlighting the percent amino acid sequence identity shared between the tail proteins of phage DMS3 and the phages indicated in the left column. Phages DMS3, JBD26 (in the absence of anti-Tab), D3112 MP29 and JBD5 are sensitive to Tab activity, whereas JBD16C, JBD88a, and MP22 are insensitive to Tab-mediated defense. **c** Ten-fold serial dilutions of the noted phages plated on cells in the absence (left) and presence (right) of Tab. Black arrow indicates very small plaques. Image with black dashed border is zoomed in to see very small plaques. Data are representative of three independent biological replicates.

the phage tail onto which the tail tube protein assembles, interfering with its function would prevent tail assembly.

## Tab defence systems include anti-Tab proteins that inhibit activity during the lytic cycle

With the knowledge that the TMP is the target of Tab, we wondered why phage JBD26 was resistant to Tab activity even though its TMP shares >99% sequence identity with the TMPs of targeted phages D3112 and MP29. Our bioinformatic analyses showed that genes encoding Tab homologues were always found between the small and large terminase genes and were accompanied by a gene encoded immediately after the small terminase gene (gene *30* in phage JBD26). In phage JBD26, the gene encoding Tab is on the minus strand and has its own promoter, allowing it to be transcribed from the prophage to provide constitutive defence. On the other hand, gene *30* is on the positive strand and overlaps the small terminase gene by one base pair, suggesting that it is transcribed on the same message with this gene and other members of the late operon.

To explore a possible functional link between Gp30 and Tab, we targeted gene *30* with a mutant *P. aeruginosa* Type I-C CRISPR-Cas system that creates short deletions[47]. While we readily isolated a phage mutant that could replicate and form plaques with a deletion that spanned both genes *30* and *tab* (Supplementary Fig. 5), deletions within gene *30* alone were not obtained. This finding suggested that production of Tab by JBD26 without Gp30, which we refer to as anti-Tab (*atab*), is lethal to the phage (i.e. prevents phage replication), and that anti-Tab protects against the inhibitory activity of Tab. The JBD26 mutant phage lacking both genes *atab* and *tab* (JBD26$^{\Delta atab/tab}$) formed plaques as efficiently as the wild-type phage, showing that these genes are not essential for phage replication (Fig. 4b). However, unlike wild-type JBD26, the JBD26$^{\Delta atab/tab}$ mutant was unable to replicate on cells

expressing Tab (Fig. 4b), implying that *atab* is required for overcoming the effects of Tab expression. To confirm this finding, we introduced the *atab* gene into the phage DMS3 genome between the small and large terminase genes, with a -1 base pair overlap with the small terminase gene, as found in JBD26 (Fig. 4a, Supplementary Fig. 5b). While wild type DMS3 is robustly inhibited by Tab activity, the DMS3$^{+atab}$ mutant was able to replicate in the presence of Tab (Fig. 4b), demonstrating that the presence of gene *atab* alone is sufficient to overcome Tab mediated inhibition. We used the bacterial adenylate cyclase-based two-hybrid (BACTH) assay[48] to determine whether anti-Tab might block Tab activity through direct interaction. We found that co-expression of Tab with anti-Tab in this system resulted in a high level of β-galactosidase activity as detected on indicator plates (Fig. 4c), suggesting a direct interaction between the two proteins. To provide additional evidence to support an interaction between these proteins, we co-expressed 6-His-tagged Tab with untagged anti-Tab in *E. coli* to determine if we could purify a complex of the two proteins. Tab expressed alone was soluble, but when co-expressed with anti-Tab, was found in the insoluble fraction of the cell (Fig. 4d). This result also suggests an interaction between the two proteins. Thus, we propose that anti-Tab binds to and inactivates Tab, thus allowing phages to escape Tab mediated inhibition of replication.

## Discussion

Here, we describe a novel prophage-encoded defence system that functions by inhibiting the tail assembly pathway of invading phages. Our data provide a straightforward mechanism for Tab anti-phage defence (Fig. 5). When JBD26 is integrated into the host cell chromosome as a prophage, the constitutive expression of Tab leaves the cell poised to defend against further phage infection. When a phage that is susceptible to Tab activity infects, its TMP is targeted by Tab activity,

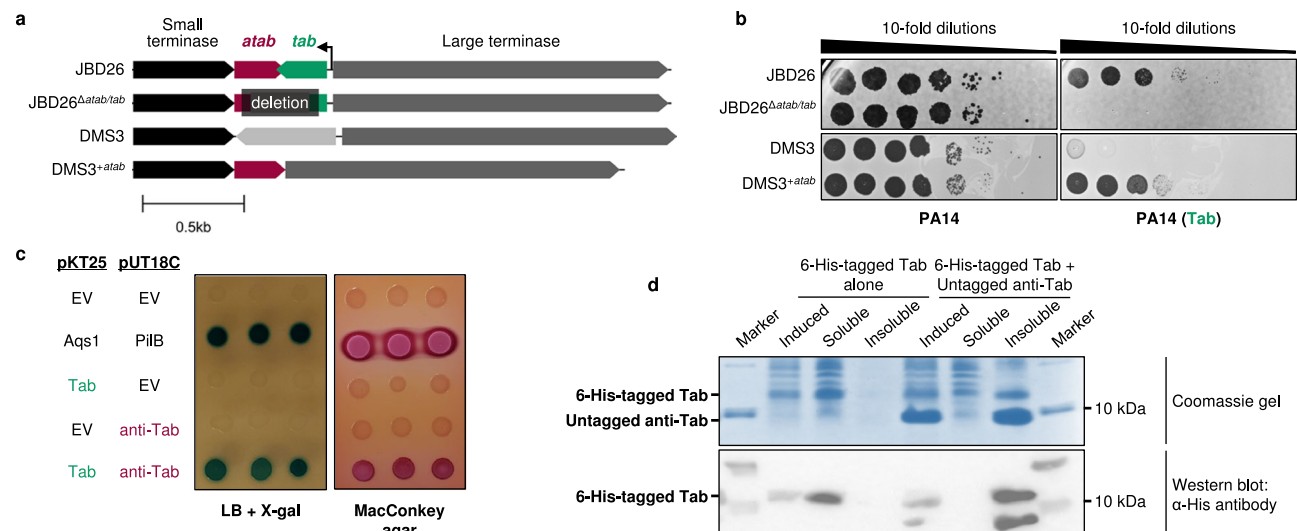

**Fig. 4 | Anti-Tab binds to Tab and inhibits its activity. a** Gene diagram highlighting differences between wild type and engineered phages in the region between the small and large terminase genes. The arrow indicates the *tab* promoter. Scale bar, 500 base pairs. **b** Ten-fold serial dilutions of the noted phages plated on lawns of PA14 carrying an empty plasmid (left) or plasmid encoding Tab (right). Data are representative of three independent biological replicates. **c** Bacterial two-hybrid assay with *E. coli* BTH101 co-transformed with different

combinations of pUT18C and pKT25 plasmids containing genes *atab* and *tab*. Positive interactions are shown by the blue color on LB plates with X-gal and red/pink color change on MacConkey plates. The Aqs1-PilB interaction serves as a positive control. **d** Analysis of soluble and insoluble fractions of the N-terminal 6-His-tagged Tab expressed alone or in combination with untagged anti-Tab by Coomassie gel (upper panel) and Western blot (lower panel). Data are representative of three independent biological replicates. Source data are provided.

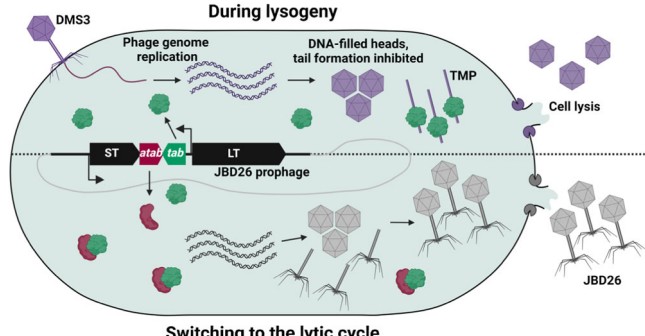

**Fig. 5 | Model for Tab-mediated defence and anti-Tab-mediated counter-defence.** During lysogeny (top), Tab is constitutively expressed from the JBD26 prophage; it detects the tape measure protein (TMP) of invading phages (purple). While DNA-filled heads and phage tail proteins are produced, tail assembly is blocked, and no mature phage particles are released. However, late-stage infection proceeds normally, leading to the accumulation of phage lysis proteins that eventually kill the cell. Since phages are not produced, secondary infections do not occur, and the bacterial community is protected. When the JBD26 prophage switches to the lytic cycle (bottom), anti-Tab is produced and binds to and sequesters Tab, allowing JBD26 replication to proceed normally. ST small terminase, LT large terminase. Created with BioRender.com.

and tail assembly is blocked. As the late genes of the invading phage, including the lysin genes, are expressed, the affected cell does not survive the infection event. However, the activity of Tab prevents the release of viable phage progeny, stopping the phage epidemic from spreading to nearby cells and protecting the bacterial community. When phage JBD26 itself enters the lytic replication cycle, expression from the late gene operon results in the production of anti-Tab, which counteracts the activity of Tab. This allows JBD26 to successfully assemble viral particles and complete the phage lifecycle.

While the precise mechanism through which Tab interferes with tail assembly is not yet known, our data allow a model to be proposed. During phage infection, the late gene operon of the invading phage

begins to produce the proteins required for assembly of the viral particle. The TMP is a key component of the phage tail, which is assembled in a highly coordinated pathway. In Mu-like phages like DMS3 and JBD26, the TMP interacts with the proteins that form the baseplate assembly at the distal tip of the tail[49]. The TMP forms a long, extended trimer or hexamer that is coated along its length by the tail assembly chaperone. The tail tube protein interacts with the tail assembly chaperone and polymerizes into hexameric rings that span the length of the TMP. The tail completion proteins then bind to the top of the tail, and the completed tail can spontaneously join with a DNA-filled head, which is assembled in a separate pathway. Our results from the TMP swapping experiments reveal that Tab-mediated inhibition is linked to a sequence in the C-terminal region of the TMP. We postulate that Tab binds to this region and either prevents the tail assembly chaperone from binding, which would in turn inhibit the assembly of the tail tube protein, or it blocks interaction of the TMP with the baseplate. While a direct interaction between Tab and the TMP is the simplest explanation, our data do not rule out an indirect effect that might lead to the observed defence. The TMP is a good target for inhibition as its activity is essential for tail formation. In addition, it is present in only three or six copies per phage, as compared to other virion proteins like the tail tube and major and minor capsid proteins which are present in hundreds of copies. Simply binding the TMP during phage assembly would be sufficient to provide defence and high amounts of the inhibitor do not need to be produced to provide robust defence against incoming phages.

As the activity of Tab inhibits phages with TMPs that share high sequence identity with phage JBD26, the prophage itself risks self-targeting. Since the defence targets the phage virion assembly pathway, the counter-defence provided by anti-Tab is only needed when the JBD26 prophage induces and enters the lytic cycle. JBD26 has ensured that anti-Tab will be produced at the right time by encoding it within the late gene operon, between the small and large terminase genes. This position allows precise temporal regulation of anti-Tab when it is needed to inactivate the constitutively produced Tab. Similar defence/counter-defence strategies have been shown in other prophage-encoded defences, such as BstA-*aba*[22]. In this system, BstA

localizes to sites of phage replication and mediates abortive infection, while the self-immunity is somehow provided through a short stretch of DNA, known as the anti-BstA element (*aba*). Detailed insight into the mechanisms of defence and counter-defence for this system remain to be elucidated. A similar defence/counter-defence activity may also exist in the phage λ RexAB system. RexB is an ion channel that triggers abortive infection upon detection of the RexA intracellular sensor[50]. It has been suggested, although not shown experimentally, that high expression of RexB might allow replication of phage λ in the presence of RexAB[51]. These types of interdependent pairs bring to mind addiction modules, as the phage-encoded defence cannot be maintained in the absence of the counter-defence.

Tab-mediated inhibition of virion assembly provides a new type of anti-phage defence. While previously identified immune systems have been shown to detect phage virion components – CBASS[33], Pycsar[52], Lit[53], PifA[54,55], SEFIR[45] and CapRel[34] recognize the major capsid protein, and DSR2[56] and Septu[45] recognize the tail tube and tail fibre, respectively – they have not been shown to inhibit viral assembly. Instead, where mechanisms are known, they have been shown to act through an active abortive infection mechanism controlled by the cell, sensing the phage structural proteins and activating an effector protein that induces growth arrest or cell suicide. Thus, while recognising phage structural components is a conserved feature of many defence systems, the direct inhibition of virion assembly is a new feature.

The Tab defence mechanism is reminiscent of the interference tactics employed by phage-inducible chromosomal islands (PICIs), which are found across a wide range of bacteria[57–59]. PICIs do not encode all of the functions necessary for virion formation but hijack the phage morphogenetic pathway of "helper" phages and co-opt their structural components for their own proliferation[60–62]. They have been shown to do this by expressing a protein that blocks the phage small terminase subunit to favour packaging of the PICI DNA or they interfere with the transcriptional regulation of the phage late genes[59,63–65]. This inhibits the assembly pathway of the helper phage, thereby decreasing phage production in order to successfully assemble PICI particles. Thus, defence is not their primary function, as it appears to be with Tab.

While Tab defence has features in common with abortive infection systems – namely, the infected cell dies and there is no release of viable phage progeny – it differs in several important ways. In contrast to many abortive infection systems[29,34,35,37,45,66,67], there is no sensing or trigger required for Tab defence; constitutive expression from the JBD26 prophage[14] leaves it poised to act on incoming phages. Additionally, destruction of the cell is not mediated through an active process associated with Tab, but by the invading phage life cycle itself. This leads to a case of mutual destruction[68], where neither the phage nor the infected cell survives. This is reminiscent of the AbiZ abortive infection system encoded by the *Lactococcus lactis* conjugative plasmid pTR2030. AbiZ activity causes phage φ31 infected cells to lyse 15 min early, leading to a 100-fold decrease in the number of infectious phage particles produced[69]. While a precise mechanism of activity is not known, AbiZ is thought to interact with the phage holin and somehow hasten lysis of the host cell. Other phage proteins that have been shown to play a role in cell death include the ToxIN[36] and DarTG[37] toxin-antitoxin systems. However, these systems are themselves toxic and will kill the cell by cleaving host transcripts or ribosylating DNA, respectively. By contrast, the activity of Tab would not kill the host cell in the absence of the phage lytic cycle. In general, any defence system that blocks phage replication at a step late in the phage life cycle is expected to share this property.

In conclusion, while homologues of Tab have only been identified in phages closely related to JBD26, small phage proteins with unknown functions are ubiquitous. As prophages are proving to be a rich source of phage defence systems, we expect other systems that block structural components in the virion assembly pathway will be uncovered;

phage structural genes are a highly conserved feature and interfering with their assembly provides a simple, yet highly effective mechanism of defence. Considering the vast and largely uncharacterized genetic diversity presented by temperate phages, prophage encoded defence and linked counter-defence systems are likely to be a highly prevalent feature.

## Methods

### Bacterial strains and phages
All bacterial strains and phages used in this manuscript are listed in Supplementary Table 1. *Pseudomonas aeruginosa* strains (UCBPP-PA14, UCBPP-PA14 ΔCRISPR[70], PA14Δ*pyoR/F* mutant[71]) and *Escherichia coli* strains (DH5α, BTH101, SM10 and BL21(λDE3)) were cultured in lysogeny broth (LB) or on LB agar. Antibiotic supplementation was used at the following concentrations when appropriate: ampicillin (100 μg/mL), kanamycin (50 μg/mL) or gentamycin (20 μg/mL) for *E. coli*; gentamycin (50 μg/mL) for *P. aeruginosa*. In *E. coli*, protein expression was induced with 1 mM isopropyl β-D-1-thiogalactopyranoside (IPTG), and 0.1% L-arabinose in *P. aeruginosa*.

### Phage preparations and plating assays
*P. aeruginosa* phages and mutant derivatives were propagated in PA14, PA14 ΔCRISPR, or PA14Δ*pyoR/F* and stored at 4 °C in SM buffer (50 mM Tris-HCl, pH 7.5, 100 mM NaCl, 8 mM MgSO$_4$, 0.01% gelatin). For phage plating assays, 150 μL of overnight bacterial culture was added to 3 mL of molten top agar (0.7%) supplemented with 10 mM Mg$_2$SO$_4$ and 0.1% L-arabinose. This mixture was poured onto LB plates with appropriate antibiotic and L-arabinose and allowed to solidify. Phage lysates were diluted in SM buffer and 2 μL aliquots of the phage dilutions were applied to the top agar overlay after it solidified. The plates were then incubated inverted overnight at 30 °C.

### Helicase attenuated Cas3 deletion of genes *30* and *31*
A type I-C CRISPR RNA (crRNA) targeting the 5′ end of gene *30* (5′–TTCTGCCGCAGCTCCAGCAGACGTACCGTCATATCCG – 3′) was cloned into a pHERD30T-derived plasmid containing modified I-C repeats[47] using primers PHP51 and PHP52, and used to transform a PAO1 strain with a chromosomally integrated Type-I-C$^{Δcsy3}$ helicase attenuated Cas3 system[47]. An overnight culture of the PAO1 I-C$^{Δcsy3}$ attenuated strain expressing the crRNA targeting gene *30* was diluted 1:100 in fresh LB supplemented with 50 μg/mL gentamycin, 0.1% L-arabinose, 0.5 mM IPTG and 10 mM MgSO$_4$ and incubated at 37 °C while shaking (250 rpm) until the OD$_{600}$ reached 0.4. Phage JBD26 was added at a multiplicity of infection (MOI) of 10, and the cultures were incubated overnight. The resulting phage lysates contained a mixed population of wild-type and mutant JBD26 phages that escaped CRISPR-Cas targeting. To select for mutants, the phage lysate was spotted on a lawn of PAO1 with the wild-type I-C CRISPR-Cas system expressing a crRNA targeting gene *30* and incubated at 30 °C. Resulting CRISPR-Cas escape mutants were plaque purified and mutations were confirmed via Sanger sequencing.

### Phage infection curves
PA14 harbouring pHERD30T or pHERD30T::*tab* was grown at 37 °C to an OD$_{600}$ of 0.2 in LB supplemented with 50 μg/mL gentamycin and 0.1% L-arabinose. 150 μL of cell culture was dispensed into the wells of a clear 96-well plate (Sarstedt TC Plate 96 well) and phages were added at the desired MOI. Bacterial growth was monitored at 37 °C with orbital shaking and measurements were taken every 30 min for 15 h using a TECAN infinite 200 plate reader. Three biological replicates were performed.

### Quantifying surviving cells post-phage infection
PA14 carrying pHERD30T or pHERD30T::*tab* was grown in LB at 37 °C to an OD$_{600}$ of 0.4. Aliquots of cells were mixed with DMS3*vir* or

DMS3*m* at an MOI of 10. After 15 min incubation, cells were collected by centrifugation, washed with LB three times to remove any unadsorbed phages, and resuspended in prewarmed LB. The cultures were incubated at 37 °C for 20 min and serial dilutions were plated on LB agar and incubated overnight at 37 °C. Colony-forming units (CFUs) were enumerated the next day. Three biological replicates were performed and mean CFUs are represented with error bars indicating the standard deviation.

## Southern blots

Overnight cultures of PA14 were diluted 1:100 in fresh LB media and incubated at 37 °C with shaking (250 rpm) to an $OD_{600}$ of 0.4. Cultures were infected with DMS3 at an MOI of 2, incubated at 37 °C, and 2 mL samples were collected at various timepoints. Cells were collected by centrifugation at 21,000 x*g* and bacterial pellets were flash frozen and stored at −80 °C. Total DNA, including both bacterial and phage DNA, was isolated from bacterial pellets as previously described[72]. Cell pellets were resuspended in 400 μL of lysis buffer (40 mM Tris-acetate, pH 7.8; 20 mM sodium acetate; 1 mM EDTA; 1% SDS) and incubated with 50 μg of RNase A at 37 °C for 30 min. NaCl was added to a final concentration of 1.25 M, and the mixture was centrifuged at 12,000 x*g* for 10 min at 4 °C. The supernatant was collected, and total DNA was isolated using phenol:chloroform:isoamyl alcohol (25:24:1, v/v; pH 7.9; Ambion) and ethanol precipitation. The purified DNA samples were digested with *Nco*I and separated on a 1% agarose gel at 150 V in TAE buffer (40 mM Tris-acetate, pH 8.3, 1 mM EDTA). DNA was transferred to a nylon membrane (Hybond-N + ; Amersham) using capillary transfer method and fixed by baking the membrane at 120 °C for 30 min. Digoxigenin (DIG-labeled) nucleic acid probes specific to DMS3 gene *25* and bacterial *rpoD* were generated by random prime labelling PCR fragments generated by primers listed in Supplementary Table 2 (primers PHP51-PHP54). DNA hybridizations were performed at 42 °C overnight using DIG Easy Hyb™ solution (Roche), and non-specifically bound probe was removed using a high stringency buffer (0.1X SSC containing 0.1% SDS; Roche) at 68 °C. DIG-labelled probes bound to DNA were detected with anti-DIG-alkaline-phosphatase antibody (1:10,000; Roche), developed using 0.25 mM CDP-*Star* solution (Roche) and imaged with Bio-Rad ChemiDoc Imaging system. Capillary transfer, probe hybridizations, probe-detection with chemiluminescence, stripping and re-probing the membrane were performed as described in Roche DIG application manual for filter hybridization using DIG-High Prime DNA Labeling and Detection Starter Kit II (Roche; 11585614910).

## Phage DNA protection assays

Bacterial pellets of phage infected cells were resuspended in ice-cold sonication lysis buffer (10 mM Tris-HCl pH 7.5, 2.5 mM MgCl$_2$, 0.5 mM CaCl$_2$) and the samples were sonicated at 30 kHz for 3 min (30 s on and 30 s off). Cell debris was removed by centrifugation at 6000 x *g* for 10 min, and the samples were incubated with DNaseI (1 μg/mL) for 1 hour at 37 °C. Following incubation, the DNase I was inactivated by adding SDS to a final concentration of 1%, and the samples were centrifuged at 12,000 x*g* for 10 min at 4 °C. Phage DNA was isolated with phenol:chloroform:isoamyl alcohol and ethanol precipitation. Finally, the DNA samples were separated on a 1% agarose gel run at 150 V in TAE buffer and Southern blots were performed as described above.

## Engineering of phage mutants

We used a two-step allelic exchange method[73] to create in-frame deletions and insertions, and to engineer genes into the DMS3 prophage. Briefly, mutant alleles containing the region of interest with flanking homology arms were amplified using PCR and were subsequently fused together using overlap-extension PCR[73,74]. The small terminase deletion (ΔST) was created with primers PHP101-PHP104; the tail tube deletion (ΔTT) with primers PHP105-108; the 6-His tag

insertion into tail tube gene with primers PHP97-PHP100; insertion of gene *30* into a DMS3 prophage with primers PHP91-PHP96; and swapping the TMP of JBD16C into a DMS3 prophage with primers PHP151-PHP156. These mutant alleles were ligated into the pEXG2 vector (Accession #: KM887143), transformed into *E. coli* SM10 strains and introduced into PA14Δ*pyoR/F* cells[71] containing a phage DMS3 prophage via conjugation. Merodiploid cells were streaked onto LB plates (no salt) with 15% (w/v) sucrose to select for cells containing double-crossover mutants. All mutant prophage sequences were confirmed by Sanger sequencing.

## Phage infection time course and Reverse-Transcription qPCR

RT-qPCR experiments were performed as previously published with minor modifications[40]. Overnight cultures were sub-cultured and grown to an $OD_{600}$ of 0.4, and phage DMS3 was added at an MOI of two. Samples were collected at 15-, 30-, 45-, and 60-min following phage challenge. RNA extraction was performed using the RNeasy Kit (Qiagen, Cat. No. 74004) and contaminating DNA was removed using Turbo DNA-free™ kit (Invitrogen, Cat. No. AM1907). RT-qPCR was performed using the Luna® Universal One-Step RT-qPCR Kit (NEB, Cat. No. E3005S). RT-qPCR primers were calibrated for quantification using DNA-standards, and standard-curves were calculated and plotted using R, and the ggplot2 and ggpmisc packages[75,76]. The data were analyzed using BioRad CFX manager 3.1 software, and relative expression data was plotted using GraphPad Prism 9.0.2. Relative expression was calculated by comparing each transcript copy number to the corresponding *rpoD* copy number. Three technical replicates were performed for each biological replicate and a total of two biological replicates were performed.

## Western blot analysis of His$_6$-tagged TT protein

Overnight cultures were diluted 1:100 in fresh LB medium and incubated at 37 °C with shaking (250 rpm) until an $OD_{600}$ of 0.4 was reached. Cultures were infected with DMS3$^{6HisTT}$ at an MOI of two. At timepoints 0, 30, and 60-min post-infection, 5 mL of cells were collected by centrifugation and flash frozen at −80 °C. The bacterial pellets were resuspended in 100 μL ice-cold sonication lysis buffer and were sonicated at 30 kHz for 5 min (30 s on, 30 s off). Cleared cell lysates were mixed with 2X Laemmli SDS loading dye (Bio-Rad), boiled at 100 °C for 20 min and then separated by 15% SDS-PAGE. For Western blot analyses, proteins on SDS-PAGE gels were transferred to a 0.45 μm nitrocellulose membrane using Bio-Rad Trans-lot SD Semi-Dry Transfer Cell at 15 V for 1 h. Western blots were incubated with primary mouse anti-His$_6$ monoclonal antibody (BioShop TAG001.100; diluted with TBS-T buffer containing Bovine serum albumin at 1:5,000) at 4 °C overnight, and then with secondary anti-Mouse IgG Horseradish peroxidase-linked antibody (Cell Signalling Technology; diluted with TBS-T buffer containing Bovine serum albumin at 1:10,000). Western blots were developed using Bio-Rad Clarity™ Western ECL substrate (Cat. # 170-5060) and imaged with the Bio-Rad ChemiDoc Imaging system.

## Attempts to isolate phage mutants that escape Tab defence

We attempted to isolate escape mutants that bypass Tab using both plating and liquid growth assays. 150 μL of PA14 expressing Tab was mixed with 150 μL of high titer phage lysate. After incubating 15 min at room temperature, 3 mL of 0.7% molten top agar was added to the culture and it was overlayed on LB agar plates. The plates were incubated at 30 °C overnight. Single plaques were isolated and serial dilutions plated on cells with and without Tab to check their ability to escape Tab defense. The entire top agar was also removed, resuspended in 3 mL of SM buffer, and replated on cells expressing Tab to enrich for phages that partially escaped Tab activity. Phages DMS3, JBD5, MP29 and JBD26$^{Δatab/tab}$ were tested, but spontaneous escape mutants were not isolated. We also carried out phage evolution

experiments[34,46,77]; PA14 cultures with and without Tab were mixed with serial dilutions of the phages noted above and were grown overnight at 37 °C in a 96-well plate. Wells showing evidence of cell lysis were collected and combined, and bacteria were removed by centrifugation. The pooled phage lysates were filtered, and then were used for subsequent rounds of infection. Additionally, after each round of infection, the pooled phage lysates were diluted and plated on cells with and without Tab to check whether phages evolved to escape Tab defence (increased EOP). Even after 20 days of phage passaging, we failed to isolate phages that could bypass Tab defence.

### Anti-Tab-Tab protein interaction assays

His$_6$-tagged Tab was cloned alone (using primers PHP160-PHP161) or in combination with untagged Anti-Tab (using primers PHP162-PHP163) into the pET-Duet-1 vector for protein co-expression and purification. These plasmids were transformed into *E. coli* BL21(λDE3) for protein expression. Overnight cultures were diluted 100-fold into fresh LB and grown to an OD$_{600}$ of 0.8. Protein expression was induced with 1 mM of IPTG, and the cultures were grown for three hours. Following the induction period, the cells were collected by centrifugation, the cell pellets were resuspended in binding buffer (20 mM Tris-HCl pH 7.5, 200 mM NaCl, and 5 mM imidazole) and the cells lysed by sonication at 30 kHz for 5 min (30 s on, 30 s off) on ice. Cell debris was collected by centrifugation at 21,000 x$g$ for 20 min at 4 °C. The insoluble fraction was resuspended in fresh binding buffer. The protein samples were mixed with 2X Laemmli SDS loading dye (Bio-Rad), boiled at 100 °C for 10 min and then separated on 15% SDS-PAGE gels before staining with Coomassie blue and subjecting to Western blot analysis as described above.

### Bacterial two-hybrid assay

The Bacterial two-hybrid (BACTH) assay was performed as described by Euromedex[48]. Tab and Anti-Tab were amplified using specific primer pairs (PHP70-PHP71 and PHP72-PHP73) and ligated into digested pKT25 and pUT18C plasmids. *E. coli* BTH101 cells were co-transformed with different combination of plasmids and plated on LB plates with ampicillin, kanamycin, 1 mM IPTG, and 40 µg/ml X-gal. The same cultures were also plated on MacConkey agar plates and incubated at 30 °C overnight. Positive interactions were indicated by a blue color change on LB plates containing X-gal and a red/pink color change on Mac-Conkey plates. Aqs1-PilB interaction was used as a positive control[26].

### Preparation of samples for electron microscopy

For the phage infection sample, PA14 overnight cultures were diluted in 1 L LB and incubated at 37 °C with constant shaking at 250 rpm until an OD$_{600}$ of 0.4. Cultures were infected with DMS3 at MOI of two. Phages were allowed to adsorb to cells for 30 min. The cultures were centrifuged to remove excess unbound phages, resuspended in fresh prewarmed LB, and incubated for three hours. PA14 lysogens of DMS3, DMS3(ΔST), DMS3(ΔTT), or a lysogen of DMS3 in PA14 expressing Tab from a plasmid, were grown in 1 L LB at 37 °C to an OD$_{600}$ of 0.4, the lysogens were induced using 1 µg/ml Mitomycin C, and were incubated for three hours, at which time cell lysis was observed. To ensure complete cell lysis, 10 mL of chloroform was added to the cultures and cell debris was removed by centrifugation at 11,000 x$g$ for 15 min. Phage lysates were treated with DNase I and RNase A to final concentrations of 1 µg/mL for 30 min at 37 °C. NaCl was added to the lysates to a final concentration of 0.5 M, and the lysates were chilled on ice for 1 h. Cell debris was removed by centrifugation, and phage lysates were filtered through a 0.22-µm filter (Millipore). The filtered lysates were treated with 10% (w/v) polyethylene glycol (PEG) 8000 (BioShop) and incubated overnight at 4 °C. Phages/phage intermediates were collected by centrifugation and resuspended in SM buffer. Cell debris and excess PEG were removed through iterative chloroform treatment and centrifugation. Final phage lysates were mixed with cesium chloride to a final concentration of 0.8 g/mL and ultracentrifuged at 229,000 x $g$ for 20 h. A high concentration of cesium chloride was used to ensure that the heads, tails, and mature phages were all present in the same fraction. After centrifugation, the 2 mL top fraction was collected and dialyzed into SM buffer.

### Transmission electron microscopy (TEM)

15 µL of phage lysate was applied to glow-discharged carbon-coated copper grids (Electron Microscopy Sciences; Carbon Film 150 Mesh, Copper) for two minutes before blotting off using a filter paper. The grids were washed three times with distilled water and stained with 2% (w/v) uranyl acetate. Imaging was performed with JEM-1011 electron microscope (JEOL USA, Inc., Peabody, MA, USA) equipped with a digital CDD camera (Model XR50, AMT Imaging, Woburn, MA, USA). All images were captured at 50,000–120,000x magnification. NucleAIzer software[78] was used for AI-based automatic segmentation of phage capsids (DNA-filled heads, empty heads or procapsids) to assess potential size differences between them. The automated segmentation picks were manually checked to ensure proper selection of heads/proheads/capsids, and ImageJ v1.54[79] was used to measure their areas. Segmentation was done for all images taken at 60,000× magnification.

### Statistical analysis

Statistical analysis was done using GraphPad Prism 9.0.2 as indicated in figure legends where necessary. The details regarding the number of biological replicates are provided in the figure legends.

### Reporting summary

Further information on research design is available in the Nature Portfolio Reporting Summary linked to this article.

## Data availability

Data that support the findings of this study are available within the article and its Supplementary Information and Source Data file. Primer sequences are available in Supplementary Table 3. Raw sequencing files associated with Fig. 3c have been uploaded as FASTA files in Supplementary Data 1. Source data are provided with this paper.

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

## Acknowledgements

We thank D. Bona for assistance with cloning and members of the Maxwell and Davidson laboratories for fruitful discussions. Phage Ab31 was generously provided by Dr. Christine Pourcel at Université Paris-Saclay. This study was supported by grants from the Canadian Institutes of Health Research to K.L.M. (PJT-165936) and A.R.D. (FDN–15427), Natural Sciences and Engineering Research Council grants to K.L.M. (RGPIN-2023-05366, SMFSU-581368-2023). K.L.M. is the Joan Dixon and Joel Parkes Professor of Biochemistry at the University of Toronto and A.R.D. is a Tier 1 Canada Research Chair in Bacteriophage-Based Technologies (950-232058).

## Author contributions

Experiments were conceived and designed by P.H.P, V.L.T., L.J.G, and K.L.M. Phage and bacterial experiments were performed by P.H.P., V.L.T., C.Z and A.D.F. RT-qPCR experiments were performed by L.J.G. A.R.D. and K.L.M. supervised the study. P.H.P. and K.L.M. wrote and revised the manuscript. All authors contributed to editing the manuscript and support the conclusions.

## Competing interests

The authors declare no competing interests.

## Additional information

**Supplementary information** The online version contains Supplementary Material available at https://doi.org/10.1038/s41467-024-45892-x.

