## [Peer Review File · Nature Communications]

Anti-phage defence through inhibition of virion assemblyEditorial Note: This manuscript has been previously reviewed at another journal that is not operating a transparent peer review scheme. This document only contains reviewer comments and rebuttal letters for versions considered at *Nature Communications*.

Reviewer #1 (Remarks to the Author):

This paper from Maxwell and colleagues has been substantially revised and improved since the original version. The authors have thoroughly responded to the initial concerns raised and addressed these concerns, with one exception. My only lingering concern are the claims that Tab directly inhibits the TMP, e.g. lines 185-6 state "This indicated that the TMP protein sequence is targeted by Tab activity," line 193 states unequivocally that "With the knowledge that the TMP is the target of Tab..." and lines 252-253 state "Our results from the TMP swapping experiments reveal that Tab recognizes a sequence in the C-terminal region of the TMP." I think it's clear that the DMS316C-TMP phage with the 16C TMP is not inhibited whereas the wt DMS3 is, and the DMS3 phage with the chimeric TMP is inhibited suggesting something important about the C-terminus of the TMP. But there are no other data to support an interaction between Tab and TMP (so the statement on lines 252-253 is especially concerning). The ability of the 16C-TMP phage to escape could stem from the alternative TMP affecting some other protein that is ultimately the direct target/binding site for Tab. This indirect effect is certainly much less likely, but it is a formal possibility in the absence of any other data, so I think the authors should better acknowledge this in the Results section and include something in the Discussion about such a limitation. I also recognize that others in the field (e.g. the Stokar-Avihail 2023 paper - ref 45) have been (over)interpreting escape mutants as proving direct triggers and targets of defense systems, but I think additional experiments are needed to rule out indirect effects, which often arise and complicate the interpretation.

This paper from Maxwell and colleagues has been substantially revised and improved since the original version. The authors have thoroughly responded to the initial concerns raised and addressed these concerns, with one exception. My only lingering concern are the claims that Tab directly inhibits the TMP, e.g. lines 185-6 state "This indicated that the TMP protein sequence is targeted by Tab activity," line 193 states unequivocally that "With the knowledge that the TMP is the target of Tab..." and lines 252-253 state "Our results from the TMP swapping experiments reveal that Tab recognizes a sequence in the C-terminal region of the TMP." I think it's clear that the DMS316C-TMP phage with the 16C TMP is not inhibited whereas the wt DMS3 is, and the DMS3 phage with the chimeric TMP is inhibited suggesting something important about the C-terminus of the TMP. But there are no other data to support an interaction between Tab and TMP (so the statement on lines 252-253 is especially concerning). The ability of the 16C-TMP phage to escape could stem from the alternative TMP affecting some other protein that is ultimately the direct target/binding site for Tab. This indirect effect is certainly much less likely, but it is a formal possibility in the absence of any other data, so I think the authors should better acknowledge this in the Results section and include something in the Discussion about such a limitation. I also recognize that others in the field (e.g. the Stokar-Avihail 2023 paper - ref 45) have been (over)interpreting escape mutants as proving direct triggers and targets of defense systems, but I think additional experiments are needed to rule out indirect effects, which often arise and complicate the interpretation.

Response:

We modified the text as follows in the results:

To identify the specific target of Tab activity, we attempted to isolate phage escape mutants using the double layer solid agar method developed by Stokar-Avihail *et al.* (2023)⁴⁵ and the liquid growth phage evolution experiments described by Srikant *et al.* (2022)⁴⁶. However, these attempts were unsuccessful, suggesting that simple mutations in the phage are not sufficient to bypass this defence, or that mutations that are able to bypass this defence are lethal to the phage. As the sequences of tail proteins of phages that were inhibited by Tab are very similar in several phages that were not affected by its activity, we performed sequence comparisons to ascertain why some phages were targeted. We noted that the proteins that comprise the tail tip (encoded by genes 42-48) generally showed high sequence conservation among phages that were both affected and unaffected by Tab activity (Fig. 3a,b). By contrast, the tail tube, tail assembly chaperone and the tape measure (TMP) proteins showed lower levels of sequence conservation, suggesting that these might be the target of Tab activity. To determine if one of these proteins was ~~responsible for~~ associated with the observed inhibition, we created a hybrid DMS3 phage where the gene encoding the TMP was replaced by the TMP gene found in phage JBD16C,

which is not blocked by Tab activity. This hybrid phage, DMS3^{16C-TMP}, formed infectious phage particles and was able to plate on cells expressing Tab, forming tiny plaques with less than a 10-fold decrease in plating efficiency, as compared to >10⁵-fold decrease observed for wild type phage DMS3 in the infection assay (Fig. 3c). This ~~indicated~~ suggested that the TMP protein sequence is targeted by Tab activity. While creating this hybrid, we isolated a second DMS3 phage that had only the first 856 residues of the TMP replaced by the JBD16C sequence. This phage, DMS3^{16C-TMP856}, was still highly susceptible to Tab mediated inhibition (Fig. 3c), demonstrating that protein sequence at the C-terminus of the TMP is ~~the target of Tab activity~~ linked with the inhibition of phage replication observed in the presence of Tab (Supplementary Fig. 4). As the TMP is a key component of the phage tail onto which the tail tube protein assembles, interfering with its function would prevent tail assembly.

We modified the text as follows in the discussion:

Our results from the TMP swapping experiments reveal that Tab-~~mediated~~ recognizes inhibition is linked to a sequence in the C-terminal region of the TMP. We postulate that Tab binds to this region and either prevents the tail assembly chaperone from binding, which would in turn inhibit the assembly of the tail tube protein, or it blocks interaction of the TMP with the baseplate. ~~While a direct interaction between Tab and the TMP is the simplest explanation, our data do not rule out an indirect effect that might lead to the observed defence.~~ The TMP is a good target for inhibition as its activity is essential for tail formation. In addition, it is present in only three or six copies per phage, as compared to other virion proteins like the tail tube and major and minor capsid proteins which are present in hundreds of copies. Simply binding the TMP during phage assembly would be sufficient to provide defence and high amounts of the inhibitor do not need to be produced to provide robust defence against incoming phages.